# The Associations between Liver Enzymes and Cardiovascular Risk Factors in Adults with Mild Dyslipidemia

**DOI:** 10.3390/jcm9041147

**Published:** 2020-04-17

**Authors:** Eun-Ock Park, Eun Ju Bae, Byung-Hyun Park, Soo-Wan Chae

**Affiliations:** 1Clinical Trial Center for Functional Foods, Chonbuk National University Hospital, Jeonju 54907, Jeonbuk, Korea; eopark@jbctc.org; 2College of Pharmacy, Chonbuk National University, Jeonju 54896, Jeonbuk, Korea; ejbae7@jbnu.ac.kr; 3Department of Biochemistry and Molecular Biology, Chonbuk National University Medical School, Jeonju 54896, Jeonbuk, Korea

**Keywords:** dyslipidemia, liver enzymes, cardiovascular risk factor, hypertension

## Abstract

Hypertension and dyslipidemia often occur as comorbidities, with both being strong risk factors for developing cardiovascular diseases (CVD). Abnormal liver function test could reflect a potential CVD risk even in patients with mild dyslipidemia. The aim of this study was to assess the compounding relationship between liver enzymes and cardiovascular risk factors in subjects with mild dyslipidemia. The present analysis was performed among 438 participants who had enrolled in at least one of the nine clinical studies done at the Chonbuk National University Hospital between 2009 and 2019. Significant linear increasing trends were observed in blood pressure level and other cardiovascular risk factors across quartiles of serum γ-glutamyltransferase (GGT) or alanine aminotransferase (ALT), with the increment in hypertension prevalence occurring across the quartiles of GGT and ALT. On multivariate logistic regression analyses, the odds ratios for hypertension, adjusted for smoking, drinking and obesity, in the highest quartiles of GGT, ALT, aspartate aminotransferase, and alkaline phosphatase were 3.688, 1.617, 1.372, and 1.166, respectively. Our study indicates that GGT is a superior marker for predicting CVD risk among liver enzymes. Routine screening of plasma GGT levels in patients with mild dyslipidemia will allow for early detection of CVD.

## 1. Introduction

Dyslipidemia and hypertension are well established risk factors for developing cardiovascular diseases (CVD). Coexistence of dyslipidemia and hypertension is often observed in clinical practice, and epidemiological studies have reported that gradual increases in blood pressure (BP) or prevalence of hypertension are associated with increases in blood lipid levels [1,2,3]. Dyslipidemia causes vascular endothelial damage [4,5,6] and impairs vasomotor activity, thereby resulting in an increase in BP, leading to hypertension. Furthermore, dyslipidemia adversely affects functional and structural arterial properties, promoting atherosclerosis [7,8,9]. Both dyslipidemia and hypertension, in turn, are commonly associated with other cardiovascular risk factors, such as type 2 diabetes and obesity [10], which further increase the risk of CVD events.

The liver plays a crucial role in the metabolism of lipids and lipoproteins through the biosynthesis of cholesterol, fatty acids, apolipoproteins, and proteins involved in lipoprotein homeostasis [11]. Liver enzymes such as γ-glutamyltransferase (GGT), alanine aminotransferase (ALT), and aspartate aminotransferase (AST) are elevated in plasma and used as markers of hepatic dysfunction due to numerous disease conditions including non-alcoholic fatty liver disease (NAFLD) [12]. In particular, GGT is a common biomarker of liver injury and alcohol consumption [13]. It is suggested that mild elevations of serum GGT and/or other liver enzymes have clinical and epidemiological significance as biomarkers of NAFLD and related liver dysfunction [14,15].

Although reports have addressed the relationship between these liver enzymes and risk factors for CVD, there exists discordance in the relationships between specific liver enzymes and CVD [4]. A meta-analysis performed by Fraser et al. [16] has shown that GGT but not ALT is associated with the incidence of coronary heart disease and stroke. Another stratified analysis has demonstrated that ALT is positively associated with stroke but negatively associated with coronary heart disease [17]. On the contrary, Lee et al. [18] showed that elevated ALT and AST are related to CVD mortality. These variations in previous findings may result from different ages, genders, ethnicities, and sample sizes of the studies and also from variable disease conditions. Nevertheless, the discovery of the association between specific liver enzymes and cardiovascular risk factors has a significant impact, especially in subjects with mild dyslipidemia, because earlier prevention or management of CVD events may have huge consequences for the individuals and society.

Mild dyslipidemia is common in the general population, particularly in subjects with elevated blood pressure. Since the co-existence of dyslipidemia and hypertension has a synergistic impact on the development of cardiovascular events and metabolic syndrome, even mild dyslipidemia needs close care and attention. Lifestyle modifications such as dietary intervention, body weight reduction, and exercise are initial steps in the treatment of mild to moderate dyslipidemia, while administration of lipid-lowering drugs is recommended in patients with frank dyslipidemia. Therefore, in the case of mild dyslipidemia, early detection and proper prevention of disease progress may be more important for preventing CVD and the associated morbidity. The aim of this study was to determine the relationship between liver enzymes and cardiovascular risk factors in adults with mild dyslipidemia.

## 2. Subjects and Methods

### 2.1. Study Population

The present study was conducted in subjects with mild dyslipidemia among participants from nine clinical trial studies performed at the Clinical Trial Center for Functional Foods (CTCF2) in Chonbuk National University Hospital between 2009 and 2019. A total of 438 participants (173 men and 265 women) with mild dyslipidemia having a complete dataset were included in our study. Written informed consent for clinical studies and for future analysis of biological specimens were obtained from all participants. Patients who met the NCEP ATP III criteria for dyslipidemia were included in the study and displayed at least one of the following three characteristics: serum low-density lipoprotein cholesterol (LDL-C) between 110 mg/dL and 190 mg/dL, total cholesterol (TC) between 200 mg/dL and 260 mg/dL, or triglyceride (TG) >150 mg/dL. The exclusion criteria in the study were as follows: having received lipid-lowering agents within the previous six months; the presence of CVD such as arrhythmia, heart failure, or myocardial infarction or usage of a pacemaker; history of alcohol or substance abuse; or the presence of other clinical diseases that may affect research results, such as uncontrolled hypertension or diabetes. The selection and exclusion of the participants for this analysis are shown in Figure 1.

### 2.2. Data Collection and Laboratory Measurements

The study protocol was approved by the Institutional Review Board of Jeonbuk National University (IRB No. JBNU 2019-12-005). For the analysis of baseline laboratory values, blood samples after an overnight fast were collected from the antecubital vein of the subjects. Patients completed a demographic information questionnaire (including current treatment, medical history, smoking status, and alcohol intake) and a health check-up according to a standardized protocol that includes electrocardiogram, laboratory tests, pulse rate, and blood pressure. Alcohol intake was calculated based on the number and type of drinks consumed by each patient and then expressed in grams of absolute alcohol per day. Routine medical examination included measurements of weight, height, body mass index (BMI), electrocardiogram, and vital signs taken in a seated position after at least 5 min of rest. Lipid profiles (TC, TG, HDL cholesterol (HDL-C), and LDL cholesterol (LDL-C)) were analyzed on a Hitachi 7600-110^®^ analyzer (Hitachi High-Technologies Corporation, Tokyo, Japan). Complete blood count was analyzed using a Sysmex XE-5000TM (Sysmex Corporation, Kobe, Japan), and other laboratory tests (liver enzymes, glucose, creatinine, etc.) were analyzed using the ADVIA^®^ 2400 chemistry system (SIEMENS, Munich, Germany).

### 2.3. Definition of Hypertension

Hypertension was defined as having a systolic BP ≥ 130 mmHg or diastolic BP ≥ 80 mmHg or use of antihypertensive medications according to the 2017 update of the American Heart Association and American College of Cardiology [19].

### 2.4. Statistical Analysis

Continuous variables are expressed as the mean with the standard deviation (SD), whereas categorical variables are expressed as the numbers and percentages. Skewed variables were logarithmically transformed to improve normality prior to analysis and then back-transformed to their natural units for presentation in the text and tables. General characteristics were analyzed by the independent *t*-test or one-way ANOVA and Chi-square test (categorized variables between the groups). The post-hoc Sheffe’s adjustment was used to compare among groups. Pearson’s correlation coefficients were used to calculate the associations of liver enzyme levels with each component of hypertension and cardiovascular risk factors. The relationship between hypertension and cardiovascular risk factors was evaluated by simple and multiple linear regression analysis. Univariate and multivariate logistic regression were performed to evaluate the odds ratios (ORs) (with 95% confidence intervals (CI)) of hypertension according to the quartiles of the liver enzyme levels. Model fitness was assessed using the Hosmer–Lemeshow test; statistically significant x^2^ statistics from this test indicate poor model fitness. The prevalence of hypertension was further verified by trend analysis. All statistical tests were two-sided, and a p-value of less than 0.05 was considered statistically significant. All statistical analyses were performed using SAS version 9.4 (SAS Institute, Cary, NC, USA).

## 3. Results

### 3.1. Characteristics of the Study Population

The baseline characteristics of study participants, who were divided into either the non-hypertensive or hypertensive group, are presented in Table 1. Among the total of 438 subjects, 247 were diagnosed with hypertension. The prevalence of hypertension was 56.39% across all subjects (28.54% were women and 27.85% were men). Subjects in the hypertensive group had significantly higher waist circumference (WC), waist hip ratio (WHR), BMI, alcohol consumption, systolic blood pressure (SBP), diastolic blood pressure (DBP), TG, very low density lipoprotein (VLDL), and TG/HDL-C but lower age and HDL-C levels than the subjects in the non-hypertensive group (Table 2). Furthermore, categorical variables such as sex, smoking, and drinking were significantly different between the groups (Table 1).

### 3.2. Changes in Cardiovascular Risk Factors along the Liver Enzyme Quartiles

Cardiovascular risk factors were analyzed according to the liver enzyme quartiles. The levels of TC, LDL-C, non-HDL-C, TC/HDL-C, and LDL-C/HDL-C rose significantly with increasing quartiles of alkaline phosphatase (ALP) (Appendix A). However, TC, LDL-C, and non-HDL-C were not different after adjusting for alcohol consumption. Similarly, GGT and ALT were also significantly associated with several lipid profiles and CVD markers (Appendix A). HDL-C was decreased, while TG, VLDL, and all CVD markers significantly increased with increasing GGT quartiles. Interestingly, SBP and DBP were positively associated with GGT quartiles, but not with ALT quartiles. Moreover, TG and TG/HDL-C increased significantly with increasing AST quartiles by trend analysis after adjusting for alcohol consumption (Appendix A). Lastly, when the association was analyzed according to gender, SBP and DBP increased with GGT quartiles both in male and in female. Other CVD markers were also positively associated with serum GGT in both sexes (Appendix A).

### 3.3. Correlation of Liver Enzymes with Cardiovascular Risk Factors

Pearson’s correlation coefficients were calculated to evaluate the correlations between liver enzyme levels and cardiovascular risk factors (Table 3). ALP was positively associated with TC, LDL-C, non HDL-C, ApoB, TC/HDL-C, LDL-C/HDL-C, ApoB/ApoA1, and hs-CRP (r = 0.117, 0.146, 0.178, 0.132, 0.181, 0.186, 0.129, and 0.147, respectively) and negatively associated with HDL-C (r = −0.119). AST was positively associated with SBP, DBP, TG, VLDL, and TG/HDL-C (r = 0.107, 0.113, 0.231, 0.185, and 0.212, respectively). ALT was positively correlated with SBP, DBP, TG, VLDL, ApoB, TC/HDL-C, LDL-C/HDL-C, TG/HDL-C, ApoB/ApoA1, and hs-CRP (r = 0.148, 0.144, 0.274, 0.196, 0.114, 0.221, 0.178, 0.294, 0.165, and 0.157, respectively) and negatively associated with HDL-C and ApoA1 (r = −0.224 and −0.128, respectively). Importantly, GGT had the highest correlation with most cardiovascular risk factors, including SBP, DBP, TG, HDL-C, non HDL-C, VLDL, ApoB, TC/HDL-C, LDL-C/HDL-C, TG/HDL-C, ApoB/ApoA1, and hs-CRP(r = 0.301, 0.335, 0.367, −0.209, 0.144, 0.295, 0.156, 0.244, 0.184, 0.340, 0.157, and 0.182, respectively).

### 3.4. Association between Liver Enzymes and Cardiovascular Risk Factors

As shown in Table 4, multiple regression analyses were performed after adjusting for alcohol consumption, age and BMI. GGT showed the highest correlation with cardiovascular risk factors compared to other liver enzymes. Unstandardized β coefficients for cardiovascular risk factors were as follows: SBP, β = 0.189; DBP, β = 0.212; TC/HDL-C, β = 0.270; LDL-C/HDL-C, β = 0.211; TG/HDL-C, β = 0.320; ApoB/ApoA1, β = 0.195; and hs-CRP, β = 0.131.

### 3.5. ORs of Hypertension along the Liver Enzyme Quartiles

Univariate and multivariate logistic regression analyses were conducted to evaluate liver enzyme levels and the risk of hypertension. In univariate logistic regression analysis, increases in the GGT quartiles were associated with increased risk of hypertension (Appendix A). Compared with subjects in the first GGT quartile, unadjusted ORs of hypertension increased from 1.581 (95% CI 0.928–2.692) for the second quartile group to 4.527 (95% CI 2.554–8.022) for the highest quartile group. After adjustment for current smoking, drinking, and obesity using the BMI, the ORs of hypertension for the highest GGT quartile was 3.688 (95% CI 1.945–6.990) (Figure 2).

### 3.6. Prevalence of Hypertension According to Liver Enzyme Quartiles by Trend Analysis

Lastly, we investigated the association between liver enzymes and the prevalence of hypertension. The prevalence of hypertension gradually increased with increases in GGT quartiles (first quartile: 40.16%, second quartile: 51.49%, third quartile: 60.91%, and fourth quartile: 75.24%; Figure 3).

## 4. Discussion

In the present study, we aimed to investigate the associations between liver enzyme activities and the risk of CVD, including measures of hypertension, in adults with mild dyslipidemia. The overall results indicated that elevated levels of liver enzymes were positively correlated with blood pressure levels and other cardiovascular risk factors, although variations were observed depending on the markers. In particular, the elevation of GGT was the most strongly associated with multiple risk factors of CVD. First of all, significant linear increasing trends were observed in the cardiovascular risk factors according to GGT quartile. Secondly, on multivariate logistic regression analyses, the OR (95% CI) for hypertension in the highest quartile of GGT, when adjusted for smoking, drinking, and obesity, was 3.688 (1.945–6.990). Lastly, the prevalence of hypertension increased from the first quartile to the highest quartile of GGT.

Our current findings of a positive correlation between CVD risks and GGT activities are consistent with previous reports. It has been observed that GGT level is positively associated with increases in ORs for all metabolic syndrome components, including central obesity, dyslipidemia, hyperglycemia, and hypertension [20]. Additionally, GGT is positively associated with the relative risk (RR) for CVD per one standard deviation change in baseline levels of liver enzymes, in which the pooled RR (95% CI) of GGT, ALP, AST, and ALT were 1.23 (1.16–1.29), 1.00 (0.99–1.02), 1.03 (0.96–1.11), and 1.08 (1.03–1.14), respectively [17]. The underlying mechanisms that link GGT with CVD risks are not fully elucidated; however, there are several possible explanations. First, GGT is regarded as a marker of oxidative stress and inflammation [20]. A longitudinal study shows clearly that GGT is positively associated with inflammation markers such as fibrinogen, hs-CRP, and F2-isoprostanes [21]. Secondly, GGT is found to play a role in the pathogenesis of atherosclerosis because it is detected in the atheromatous plaques of carotid and coronary arteries, triggering the oxidation of LDLs [22,23]. Lastly, GGT plays a role in glutathione homeostasis, which is central to antioxidant defense [21,24,25]. As such, elevated GGT levels could be a marker of inflammation and oxidative stress, which are important features of CVD.

Dyslipidemia is highly prevalent in children with NAFLD, and serum TG and non-HDL-C may play a critical role in the pathogenesis of NAFLD [26]. Routine evaluation of blood lipid profiles and the subsequent management of dyslipidemia are warranted in NAFLD patients. The role of dyslipidemia in the development and progression of NAFLD has been extensively studied, but the underlying mechanisms that link liver enzymes with CVD are not well established [27,28]. Disturbances in hepatic lipid metabolism may influence the development of NAFLD in the presence of metabolic derangements such as obesity, insulin resistance, hyperglycemia, and hypertension. Moreover, NAFLD is associated with vascular inflammation, which may reflect vulnerable rupture-prone atherosclerotic plaques [29]. In the current study, we found that cardiovascular risk factors were associated with elevated liver enzymes, especially GGT and ALT. The presence and severity of NAFLD may bridge an increased risk of dyslipidemia and CVD.

Our findings of the relationships between liver enzymes and cardiovascular risk factors are supported by previous studies. Indeed, both visceral fat accumulation and risk of metabolic syndrome are significantly correlated with GGT and ALT levels [30,31]. A meta-analysis shows that GGT levels are positively and independently associated with cardiovascular risks [17,32,33]. There is a monotonous increase in CVD risk that is proportional to increasing levels of GGT. Importantly, a recent nationwide population-based cohort study performed in a Korean population has demonstrated that, among liver enzymes, GGT was the most strongly implicated in the development of CVD and all-cause mortality [34], keeping in line with our current finding. Altogether, apart from liver diseases, accumulating evidence has shown that GGT is a risk indicator for hypertension [21,35,36,37,38].

Dyslipidemia requires lifelong treatment with regular follow-up to reduce the risk of CVD [39,40], and an earlier diagnostic cascade screening strategy is important because earlier treatment is highly beneficial. Generally, patients aged ≥ 21 years with LDL-C ≥ 190 mg/dL (not due to modifiable secondary causes), complex mixed dyslipidemias, or severe hypertriglyceridemia may consider combining drug therapy. On the other hand, in cases of mild dyslipidemia, lifestyle modifications are necessary to prevent disease progress and CVD [39]. Lifestyle changes include dietary modification, regular exercise, maintenance of healthy body weight, reduction of alcohol consumption, and avoidance of smoking. Since cardiovascular risk is not found to be linear with serum lipid levels in mild dyslipidemia, there are potential advantages to utilizing liver enzymes to predict CVD because liver enzyme tests are a low-cost, simple, sensitive, and standardized method [17].

This study has several limitations. First, this study has variations in data collection and potential selection biases. In addition, as the study analysis was performed in a single clinical trial research center and the sample size is relatively small, the results may not be generalizable to the full population. Further study is warranted involving the larger sample population. Second, elevations in serum liver enzyme activities cannot differentiate a variety of liver diseases, such as acute hepatitis, chronic liver diseases, and biliary diseases. Third, various inflammatory markers associated with CVDs were not measured. Fourth, chronic heavy drinkers are not excluded from the study, although liver enzyme level is a sensitive indicator of alcohol intake. Lastly, due to the small number of sample size and the feature of mild dyslipidemia, we could not evaluate whether liver enzymes or the associated CV risk factors also implicated in major adverse cardiovascular events (MACE) such as myocardial infarction and ischemic stroke or mortality.

Despite these limitations, this study is the first, to our knowledge, to investigate the relationship between GGT levels and CVD risk factors, specifically in adults with mild dyslipidemia. In conclusion, our study suggests that serum GGT level is independently associated with cardiovascular risk factors and proposes its potential prognostic value for CVD in people with mild dyslipidemia. Further studies are needed to expand our findings and to demonstrate a relationship with GGT/CV risk and MACE.

## Figures and Tables

**Figure 1 jcm-09-01147-f001:**
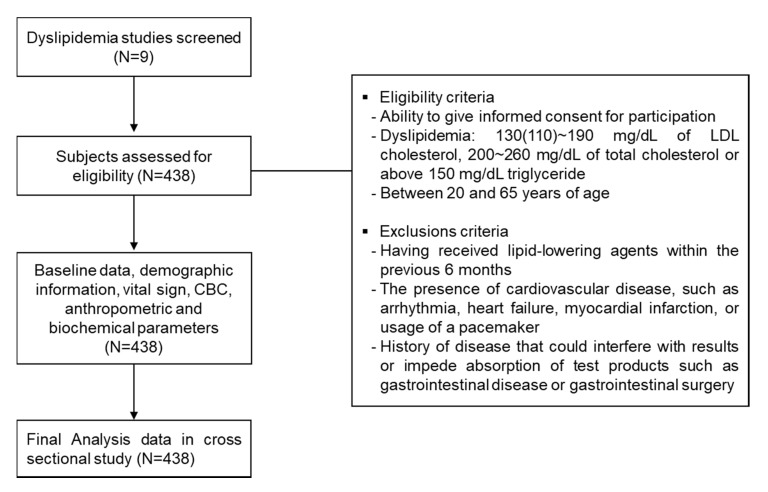
Flow diagram of study selection process.

**Figure 2 jcm-09-01147-f002:**
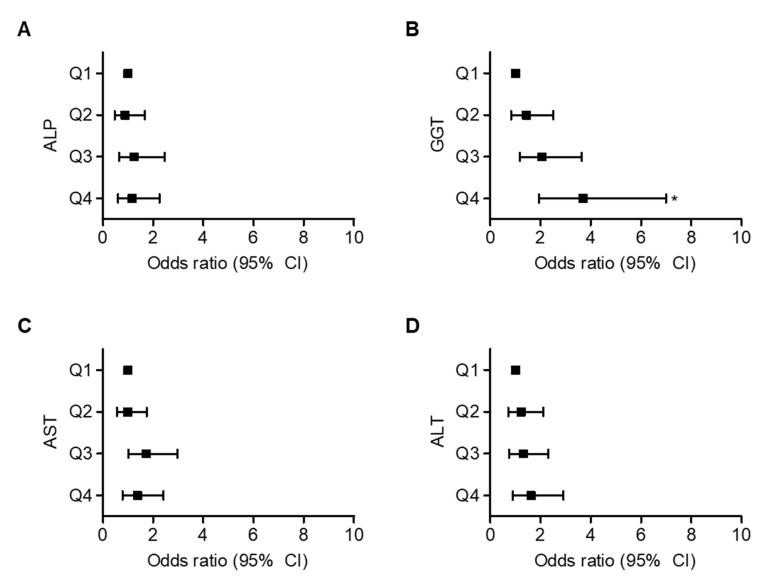
Odds ratios (ORs) with 95% CI for hypertension stratified by smoking drinking and obesity adjusted liver enzymes quartile. The quartile with the lowest liver enzymes is used as reference. Q1 indicates first quartile and so on. (**A**) ALP, *p* = 0.749; (**B**) GGT, *p* = 0.001; (**C**) AST, *p* = 0.154; (**D**) ALT, *p* = 0.452. * *p* < 0.05 vs. Q1. Analyzed by Logistic regression analysis. Hosmer and Lemeshow’s goodness of fit test (x^2^ > 0.05). Abbreviation: ALP, alkaline phosphatase; GGT, gamma-glutamyltransferase; AST, aspartate aminotransferase; ALT, alanine aminotransferase; OR, odds ratio; Cl, confidence interval.

**Figure 3 jcm-09-01147-f003:**
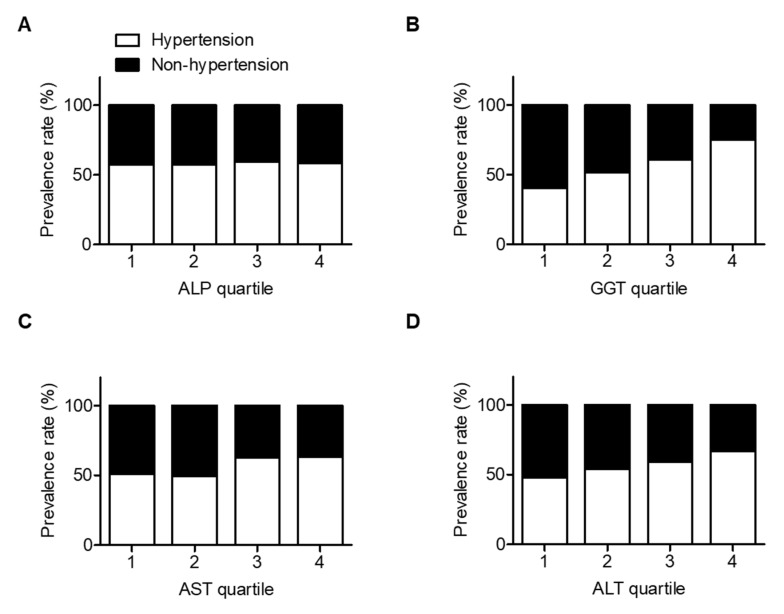
Prevalence of hypertension rate (%) according to liver enzymes quartile. (**A**) ALP, *p* for trend = 0.846; (**B**) GGT, *p* for trend <0.001; (**C**) AST, *p* for trend = 0.018; and (**D**) ALT, *p* for trend = 0.004. Data are presented as N (%), Analyzed by Cochran–Armitage trend analysis. Abbreviation: ALP, alkaline phosphatase; GGT, gamma-glutamyltransferase; AST, aspartate aminotransferase; ALT, alanine aminotransferase.

**Table 1 jcm-09-01147-t001:** General characteristics.

		Non-Hypertension (n = 191)			Hypertension (n = 247)			*p*-Value ^a^
Sex	Female	140 (31.96)			125 (28.54)			<0.001 ^b^
	Male	51 (11.64)			122 (27.85)			
Age (years)		50.51	±	9.02	48.53	±	9.51	0.027
Height (cm)		161.53	±	8.20	164.41	±	9.00	0.001
Weight (kg)		64.85	±	11.11	70.76	±	13.17	<0.001
BMI (kg/m^2^)		24.76	±	3.21	26.03	±	3.23	<0.001
WC (cm)		86.26	±	8.85	88.89	±	7.16	0.034
HC (cm)		95.63	±	5.59	96.77	±	6.17	0.218
WHR		0.90	±	0.06	0.92	±	0.05	0.026
Antihypertensive drug, N (%)		0 (0)			20 (4.57)			<0.001 ^b^
Smoker		15 (3.42)			36 (8.22)			0.049 ^b^
Drinker		69 (15.75)			133 (30.37)			0.001 ^b^
Alcohol consumption ^†^ (g/week)		3.12	±	2.93	6.88	±	8.44	<0.001
Food consumption (kcal)		1606.19	±	420.56	1657.09	±	430.10	0.261
Physical activity ^†^ (METs/week)		1897.20	±	2387.13	2244.61	±	3104.49	0.316

Abbreviation: BMI, body mass index; WC, waist circumference; HC, hip circumference; WHR, waist hip ratio. Values are presented as mean ± SD or number (%). † Log transformed for comparisons to their natural units for presentation. a, Analyzed by independent *t*-test. b, Analyzed by χ^2^ test.

**Table 2 jcm-09-01147-t002:** Cardiovascular risk factors.

	Non-Hypertension (n = 191)			Hypertension (n = 247)			*p*-Value ^a^
SBP (mmHg)	112.15	±	8.84	131.47	±	9.72	<0.001
DBP (mmHg)	70.88	±	5.56	87.19	±	6.94	<0.001
TC (mg/dL)	218.54	±	26.85	214.53	±	24.03	0.101
TG (mg/dL)	146.64	±	59.58	163.20	±	69.18	0.008
HDL-C (mg/dL)	51.66	±	12.23	50.46	±	11.25	0.288
LDL-C (mg/dL)	142.29	±	24.95	135.50	±	22.87	0.003
Non HDL-C (mg/dL)	166.88	±	26.15	164.07	±	22.55	0.237
VLDL (mg/dL)	32.06	±	7.81	35.55	±	9.19	0.019
ApoA1 (g/L)	1.45	±	0.23	1.45	±	0.25	0.864
ApoB (g/L)	1.22	±	0.21	1.21	±	0.19	0.346
TC/HDL-C	4.41	±	0.96	4.40	±	0.84	0.922
LDL-C/HDL-C	2.89	±	0.79	2.80	±	0.68	0.207
TG/HDL-C	3.11	±	1.72	3.49	±	1.89	0.033
Apo B/ApoA1	0.87	±	0.22	0.86	±	0.20	0.667
hs-CRP ^†^ (mg/L)	0.92	±	2.03	0.84	±	1.68	0.168

Abbreviation: SBP, systolic blood pressure; DBP, diastolic blood pressure; TC, total cholesterol; TG, triglyceride; HDL-C, high-density lipoprotein cholesterol; LDL-C, low-density lipoprotein cholesterol; VLDL, very low density lipoprotein; ApoA1, apolipoprotein A1; ApoB, apolipoprotein B; hs-CRP, high sensitivity C-reactive protein. Values are presented as mean ± SD. † Log transformed for comparisons to their natural units for presentation. a, Analyzed by independent *t*-test.

**Table 3 jcm-09-01147-t003:** Correlation of liver enzymes with cardiovascular risk factors.

	ALP		GGT		AST		ALT	
	*r*	*p*-Value	*r*	*p*-Value	*r*	*p*-Value	*r*	*p*-Value
SBP	0.089	0.104	0.301	<0.001	0.107	0.025	0.148	0.002
DBP	0.001	0.991	0.335	<0.001	0.113	0.018	0.144	0.002
TC	0.117	0.032	0.042	0.386	−0.052	0.281	−0.047	0.325
TG	0.062	0.258	0.367	<0.001	0.231	<0.001	0.274	<0.001
HDL-C	−0.119	0.029	−0.209	<0.001	−0.060	0.208	−0.224	<0.001
LDL-C	0.146	0.007	0.012	0.804	−0.070	0.145	−0.018	0.700
Non HDL-C	0.178	0.001	0.144	0.003	−0.025	0.603	0.059	0.220
VLDL	0.149	0.083	0.295	0.001	0.185	0.031	0.196	0.022
ApoA1	−0.079	0.165	-0.056	0.262	0.039	0.433	−0.128	0.010
ApoB	0.132	0.021	0.156	0.002	0.052	0.301	0.114	0.022
TC/HDL-C	0.181	0.001	0.244	<0.001	0.039	0.412	0.221	<0.001
LDL-C/HDL-C	0.186	0.001	0.184	0.0001	0.005	0.922	0.178	0.002
TG/HDL-C	0.083	0.126	0.340	<0.001	0.212	<0.001	0.294	<0.001
ApoB/ApoA1	0.129	0.024	0.157	0.002	0.013	0.794	0.165	0.001
hs-CRP ^†^	0.147	0.019	0.182	0.001	0.073	0.171	0.157	0.003

Abbreviation: ALP, alkaline phosphatase; GGT, gamma-glutamyltransferase; AST, aspartate aminotransferase; ALT, alanine aminotransferase; SBP, systolic blood pressure; DBP, diastolic blood pressure; TC, total cholesterol; TG, triglyceride; HDL-C, high-density lipoprotein cholesterol; LDL-C, low-density lipoprotein cholesterol; VLDL, very low density lipoprotein; ApoA1, apolipoprotein A1; ApoB, apolipoprotein B; hs-CRP, high sensitivity C-reactive protein. Pearson’s correlation coefficients. † Log transformed for comparisons to their natural units for presentation.

**Table 4 jcm-09-01147-t004:** Multivariate regression analysis of liver enzymes with cardiovascular disease (CVD) risk factors.

	ALP			GGT			AST			ALT	
	*β*	*adj. R* ^2^	*p*-Value	*β*	*adj. R* ^2^	*p*-Value	*β*	*adj. R* ^2^	*p*-Value	*adj. R* ^2^	*p*-Value
SBP	0.125	0.107	0.023	0.189	0.148	0.0003	0.024	0.122	0.614	0.122	0.530
DBP	0.032	0.118	0.561	0.212	0.167	<0.001	0.017	0.133	0.721	0.133	0.550
TC	0.089	0.045	0.117	0.123	0.049	0.026	−0.050	0.040	0.317	0.038	0.967
TG	0.076	0.054	0.181	0.340	0.148	<0.001	0.182	0.091	0.0002	0.106	<0.001
HDL-C	−0.099	0.077	0.076	−0.189	0.118	0.0004	−0.036	0.092	0.461	0.106	0.008
LDL-C	0.104	0.049	0.066	0.084	0.044	0.128	−0.068	0.043	0.173	0.039	0.970
Non HDL-C	0.138	0.059	0.014	0.220	0.069	<0.001	−0.035	0.033	0.488	0.036	0.199
VLDL	0.211	0.054	0.018	0.264	0.068	0.006	0.165	0.040	0.055	0.051	0.023
Apo A1	−0.036	0.088	0.540	-0.057	0.080	0.310	0.041	0.079	0.414	0.079	0.404
Apo B	0.096	0.062	0.109	0.226	0.089	<0.001	0.028	0.051	0.591	0.061	0.030
TC/HDL-C	0.138	0.083	0.013	0.270	0.131	<0.001	0.017	0.076	0.729	0.096	0.003
LDL-C/HDL-C	0.138	0.101	0.013	0.211	0.114	<0.001	−0.016	0.080	0.746	0.091	0.027
TG/HDL-C	0.079	0.036	0.165	0.320	0.140	<0.001	0.175	0.091	0.0003	0.110	<0.001
Apo B/ApoA1	0.080	0.067	0.184	0.195	0.084	0.001	−0.001	0.054	0.883	0.063	0.055
hs-CRP ^†^	0.110	0.062	0.082	0.131	0.084	0.027	0.030	0.072	0.574	0.074	0.282

Abbreviation: ALP, alkaline phosphatase; SBP, systolic blood pressure; DBP, diastolic blood pressure; TC, total cholesterol; TG, triglyceride; HDL-C, high-density lipoprotein cholesterol; LDL-C, low-density lipoprotein cholesterol; VLDL, very low density lipoprotein; Apo A1, apolipoprotein A1; Apo B, apolipoprotein B; hs-CRP, high sensitivity C-reactive protein. † Log transformed for comparisons to their natural units for presentation. β = unstandardized regression coefficient; adj. R^2^ = a measure for the model prediction. Multiple regression analysis is adjusted for alcohol consumption, age, and BMI.

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
