# Peer review of "The Associations between Liver Enzymes and Cardiovascular Risk Factors in Adults with Mild Dyslipidemia"

_jcm, 2020, doi:10.3390/jcm9041147_

Round 1
Reviewer 1 Report
I believe this manuscript represents an area of recent interest and investigation that has thus far shown some potential value in identifying another biomarker for CV risk. This paper adds to existing literature on this topic, and does assist in clarifying some conflicting results to date. I believe this paper could be improved upon by adding some discussion or clarification related to the following points:
1. Overall the sample size of this analysis is small, as the authors have acknowledged. Notably, the non-hypertension group is disproportionately smaller and under-represented by males compared to the hypertension group. The authors state ""results may not be generalizable to the full population." Which population would they be applied to if this is true?
2. Other studies that have examined the association between GGT and CV risk have mentioned the possibility of gender/ethnicity variations. Are the authors able to exclude this interaction from impacting the results presented here?
3. The study acknowledges the potential role of alcohol consumption as a modifier of liver enzymes and CV risk, but have not mentioned any consideration of the role of other medications used at baseline. Were baseline medications (other than lipid lowering medications) that might influence lipid status, liver enzymes or CV risk accounted for? (hormone replacement therapies, oral contraceptives, protease inhibitors, retinoids, systemic steroid use, antihypertensives (beta blockers, thiazide diuretics), etc. considered?
4. Please clarify if the biomarkers obtained represent a single lab draw in isolation. (only 1 set of biomarkers drawn) If so, we must assume that 1 lab draw of biomarkers is representative of the subjects chronic health status and not an outlier.
5. Finally, data presented here are derived from 9 clinical trial studies. The authors are clear in their study aim to determine the relationship between liver enzymes and CV risk factors. However, the ultimate value of this information is to demonstrate that the CV risk factor is associated with a CV clinical event. The authors have excluded patients who have existing CVD, but to be proven useful as a CV risk factor, we need to determine if major adverse clinical events occur. Please comment on this need. At a minimum, a suggestion that further studies are needed to demonstrate a relationship with GGT/CV risk and MACE's?
Author Response
I believe this manuscript represents an area of recent interest and investigation that has thus far shown some potential value in identifying another biomarker for CV risk. This paper adds to existing literature on this topic, and does assist in clarifying some conflicting results to date. I believe this paper could be improved upon by adding some discussion or clarification related to the following points:
- Overall the sample size of this analysis is small, as the authors have acknowledged. Notably, the non-hypertension group is disproportionately smaller and under-represented by males compared to the hypertension group. The authors state ""results may not be generalizable to the full population." Which population would they be applied to if this is true?
--> As commented, we thought the small number of sample size was one of the limits of this study particularly in the male group due to the disproportionate distribution between non-hypertension and hypertension group. Relating to the Reviewer’s comment 2, we now further analyzed data and found the positive association between GGT and CV risks both in the male and female groups. Nonetheless, further study involving the larger sample population is needed and we added this point in the Discussion section as follows:
Further study is warranted involving the larger sample population.
- Other studies that have examined the association between GGT and CV risk have mentioned the possibility of gender/ethnicity variations. Are the authors able to exclude this interaction from impacting the results presented here?
--> This study was performed in Korean people, a single ethnic group. For gender variation, we added the association between GGT and CV risks according to gender in Tables S5 and S6 in the revised manuscript and described the results as follows:
Lastly, when the association was analyzed according to gender, SBP and DBP increased with GGT quartiles both in male and in female. Other CVD markers were also positively associated with serum GGT in both sexes (Tables S5 and S6).
- The study acknowledges the potential role of alcohol consumption as a modifier of liver enzymes and CV risk, but have not mentioned any consideration of the role of other medications used at baseline. Were baseline medications (other than lipid lowering medications) that might influence lipid status, liver enzymes or CV risk accounted for? (hormone replacement therapies, oral contraceptives, protease inhibitors, retinoids, systemic steroid use, antihypertensives (beta blockers, thiazide diuretics), etc. considered?
--> We thank the Reviewer for this comment. As we described in the exclusion criteria, subjects under lipid lowering medications were excluded in this study (in line 78 and Figure 1). Subjects taking other medications that could affect lipid metabolism were also excluded in the study and the individuals taking antihypertensive drug for BP control (4.57 % of participants) were included. We added this point in Table 1 in the revised manuscript.
- Please clarify if the biomarkers obtained represent a single lab draw in isolation. (only 1 set of biomarkers drawn) If so, we must assume that 1 lab draw of biomarkers is representative of the subjects chronic health status and not an outlier.
--> As commented, the biomarkers obtained in this study represent a single lab draw in isolation. We instructed the study participants to maintain their usual dietary habits and to refrain from alcohol drinking, high-fat food, excessive fasting, etc., to reflect their chronic health conditions.
- Finally, data presented here are derived from 9 clinical trial studies. The authors are clear in their study aim to determine the relationship between liver enzymes and CV risk factors. However, the ultimate value of this information is to demonstrate that the CV risk factor is associated with a CV clinical event. The authors have excluded patients who have existing CVD, but to be proven useful as a CV risk factor, we need to determine if major adverse clinical events occur. Please comment on this need. At a minimum, a suggestion that further studies are needed to demonstrate a relationship with GGT/CV risk and MACE's?
--> We agree on this with the Reviewer. In the revised manuscript, we commented this point as follows.
Lastly, due to the small number of sample size and the feature of mild dyslipidemia, we could not evaluate whether liver enzymes or the associated CV risk factors also implicated in major adverse cardiovascular events (MACE) such as myocardial infarction and ischemic stroke or mortality.
…
Further studies are needed to expand our findings and to demonstrate a relationship with GGT/CV risk and MACE.
Reviewer 2 Report
Authors have performed a well-structured study in 438 patients with mild dyslipidemia (173 men/265 women) to test the possible correlation between liver enzymes levels and some cardiovascular risk factors. As authors include in their bibliography (Kunutsor SK; Atherosclerosis. 2014), a wide range of similar studies evaluating liver enzymes and cardiovascular disease risk in general populations have been published previously. However, as authors highlight in their conclusions, this is the first study evaluating a correlation between liver enzymes levels and cardiovascular risk factors in adults with mild dyslipidemia. There are limitations in the study that have been also well explained by authors. Finally, authors conclude that their study “suggests that serum GGT level is independently associated with cardiovascular risk factors and proposes its potential prognostic value for CVD in people with mild dyslipidemia”. Although it is not a very novel study it is well designed and analyzed, and results could help in future studies. I recommend to revise and include some recent articles about this topic such us Choi KM et al. 2018 (PMID: 29491346)
Author Response
Authors have performed a well-structured study in 438 patients with mild dyslipidemia (173 men/265 women) to test the possible correlation between liver enzymes levels and some cardiovascular risk factors. As authors include in their bibliography (Kunutsor SK; Atherosclerosis. 2014), a wide range of similar studies evaluating liver enzymes and cardiovascular disease risk in general populations have been published previously. However, as authors highlight in their conclusions, this is the first study evaluating a correlation between liver enzymes levels and cardiovascular risk factors in adults with mild dyslipidemia. There are limitations in the study that have been also well explained by authors. Finally, authors conclude that their study “suggests that serum GGT level is independently associated with cardiovascular risk factors and proposes its potential prognostic value for CVD in people with mild dyslipidemia”. Although it is not a very novel study it is well designed and analyzed, and results could help in future studies. I recommend to revise and include some recent articles about this topic such us Choi KM et al. 2018 (PMID: 29491346)
--> As the Reviewer commented, we revised the discussion as follows and added a new recent reference.
Importantly, a recent nationwide population-based cohort study performed in Korean population has demonstrated that, among liver enzymes, GGT was the most strongly implicated in the development of CVD and all-cause mortality [34], keeping in line with our current finding.
Reviewer 3 Report
The article, "The associations between liver enzymes and cardiovascular risk factors in adults with mild dyslipidemia" describes an important observation that will have immediate clinical and therapeutic implications in patients with mild hypertension. The study was designed with precision and several statistical analysis were done accurately.
Please indicate how it was ensured that participants in an individual clinical trial study at the same Center (CTCF) at the Chonbuk National University Hospital were not included in multiple studies at the same Center?
The authors should consider confusing the word "concentrations" in lines 205, 214 and 258, as the determination of liver enzyme "activity" is performed.
Author Response
The article, "The associations between liver enzymes and cardiovascular risk factors in adults with mild dyslipidemia" describes an important observation that will have immediate clinical and therapeutic implications in patients with mild hypertension. The study was designed with precision and several statistical analysis were done accurately.
Please indicate how it was ensured that participants in an individual clinical trial study at the same Center (CTCF) at the Chonbuk National University Hospital were not included in multiple studies at the same Center?
--> Patient data is protected by personal identification numbers so it is not possible to check whether or not an individual has participated in multiple studies.
The authors should consider confusing the word "concentrations" in lines 205, 214 and 258, as the determination of liver enzyme "activity" is performed.
--> We thank the Reviewer for the correction. As suggested, we replaced the word liver enzyme “concentrations" to "activities".